# Rectification and confinement of photokinetic bacteria in an optical feedback loop

Helena Massana-Cid [1], Claudio Maggi [1,2], Giacomo Frangipane [1,3] & Roberto Di Leonardo [1,2✉]

Active particles can self-propel by exploiting locally available energy resources. When powered by light, these resources can be distributed with high resolution allowing spatio-temporal modulation of motility. Here we show that the random walks of light-driven bacteria are rectified when they swim in a structured light field that is obtained by a simple geometric transformation of a previous system snapshot. The obtained currents achieve an optimal value that we establish by general theoretical arguments. This optical feedback is used to gather and confine bacteria in high-density and high-activity regions that can be dynamically relocated and reconfigured. Moving away from the boundaries of these optically confined states, the density decays to zero in a few tens of micrometers, exhibiting steep exponential tails that suppress cell escape and ensure long-term stability. Our method is general and scalable, providing a versatile tool to produce localized and tunable active baths for micro-engineering applications and systematic studies of non-equilibrium phenomena in active systems.

[1] Dipartimento di Fisica, Sapienza Università di Roma, Piazzale A. Moro 5, I-00185 Rome, Italy. [2] NANOTEC-CNR, Soft and Living Matter Laboratory, Institute of Nanotechnology, Piazzale A. Moro 5, I-00185 Rome, Italy. [3] Center for Life Nano & Neuroscience, Fondazione Istituto Italiano di Tecnologia (IIT), 00161 Rome, Italy. ✉email: roberto.dileonardo@uniroma1.it

The possibility of having a light-controllable self-propulsion speed at the microscale has been recently demonstrated in both synthetic active particles[1–4] and motile biological cells[5]. This, coupled with spatial light modulators, allows the study of active matter with motility characteristics that can be precisely controlled in space and time. For instance, stationary distributions of non interacting active particles with a space dependent speed are characterized by a constant value of the product $\rho v$ between local values of density $\rho$ and velocity modulus $v$[6,7]. This has been exploited in light driven bacteria to paint reconfigurable structures using static light patterns, where cells accumulate in darker and thus slower regions[8,9]. Static light patterns, however, cannot be used to generate stationary currents in systems of purely photokinetic particles because of the time reversal symmetry of microscopic dynamics[10]. Furthermore, the study of photokinetic cells in static patterns is often limited in time because of continuous escape of motile cells outside the observation area. This is because structured illumination is confined by the finite size of the spatial light modulator, so that the system is surrounded by a dark region of particles with $v = 0$. Since the product $\rho v$ must remain approximately constant throughout the system, bacteria will over time be absorbed by the low-motility surroundings. To avoid this we must develop new light control strategies to generate stationary currents that counterbalance the leaking of active particles outside of the illumination area. Methods to achieve a net current in light-driven systems range from taking advantage of an orientational response to light intensity gradients in a system of synthetic photoactivated microparticles[11,12] to breaking time reversal symmetry at a microscopic level[13] by projecting dynamic light patterns with a time asymmetry[10,14,15]. Another possible strategy is to break time reversal symmetry through particle interactions as proposed in simulations[16]. None of these methods, however, combines the efficiency and scalability that are required to accumulate large numbers of active particles in regions of high density and high motility.

Here we demonstrate that, by projecting back onto a suspension of photokinetic bacteria a properly transformed image of the system at an earlier time, we can guide cells with maximum efficiency along a designable flow field. Specifically, an optical feedback keeps the cells under a binary illumination pattern such that only bacteria moving in the desired direction are illuminated by a bright green spotlight and swim fast. Other cells are kept under a background of low light intensity and swim slower until a tumble turns them in the right direction. We exploit this method to optically gather and confine large collections of cells in high density and high activity regions over long time periods. The spatial structure of these confined stationary states is quantitatively described by a run and tumble model of active particles with a self propulsion speed that depends on both position and direction. Furthermore, we prove that these active formations can be split, merged and relocated in space, while conserving the total number of trapped cells. In a broader context, controllable currents of active particles[17,18] may have important applications for the transport of passive cargoes in microfluidic environments[19–21], for microengineering[22–24], and for fundamental studies in the statistical physics of active systems[25–29].

## Results

**A Maxwell demon for active particles.** Let us suppose that one has complete knowledge of the individual positions and velocities of active particles. Such a "Maxwell demon"[30] could use this information to generate a net flow in a desired direction $\hat{\mathbf{n}}$ by adjusting individual particles' speeds. The maximal current that the demon could produce in this way would be obtained by giving the maximum available speed $v_1$ to all particles with a positive velocity component along $\hat{\mathbf{n}}$, and the minimum available speed $v_0$

to those moving in the opposite direction:

$$v(\theta) = \begin{cases} v_1 & \text{if } |\theta| < \pi/2 \\ v_0 & \text{if } |\theta| > \pi/2 \end{cases} \quad (1)$$

where $\theta$ is the angle that the particle velocity forms with $\hat{\mathbf{n}}$. In a 2D system, with an isotropic and homogeneous reorientational dynamics, the average velocity component in the $\hat{\mathbf{n}}$ direction would have a magnitude of:

$$\bar{v}^* = \frac{1}{2\pi} \int_0^{2\pi} v(\theta) \cos\theta \, d\theta = \frac{v_1 - v_0}{\pi} \quad (2)$$

In photokinetic bacteria, swimming speed is a function of incident light so that individual speeds can be controlled by projecting tailored light patterns. Although feasible in principle, the control strategy described above would require continuous particle tracking which may make it difficult to implement and not easily scalable to large systems. We devise a much simpler, faster and scalable strategy consisting of three main steps. Using dark-field optical microscopy, we first capture a snapshot of the system in which bacteria appear as white spots on a dark background. We then dilate the image using a circular kernel and apply a threshold so that every cell in the original image is replaced by an approximately circular blob of average radius $R$. Then we remap this image by a plane transformation that moves the content of every pixel by a distance $\Delta$ towards the desired flow direction $\hat{\mathbf{n}}$ and finally we project it back onto the system after a delay time $\tau$ (Fig. 1a). In this way, bacteria that are moving in the positive $\hat{\mathbf{n}}$ direction would be constantly under the spotlight and swim fast while bacteria moving in the opposite direction would be in the dark and swim slow. Like a "Maxwell demon", the feedback loop rectifies the motion of bacteria by "opening or closing the door", i.e. bright or dark exposure, depending on the direction of motion of the particles.

To implement the method experimentally, we fill a 10 μm height chamber with genetically modified *E. coli* bacteria expressing the light-driven proton pump proteorhodopsin (see Methods). The chamber is sealed so that, after the cells have consumed all the oxygen in the buffer through respiration, the proton motive force drops down and the flagellar motors stop spinning and start responding to external green light stimuli[5]. Using a digital light projector, we can spatially control swimming speeds by illuminating the sample with green light patterns. In particular, using the feedback algorithm described above, we generate a binary image with bright spots of intensity $I_1 = 4.6$ mW mm$^{-2}$ over a background of intensity $I_0 = 0.2$ mW mm$^{-2}$. As shown in Fig. 1b, when we apply a shift in the positive $x$ direction (axial transformation in Fig. 1a), bacteria that swim towards the right move faster than those swimming to the left. This is evidenced in the figure by the white traces obtained by integrating images acquired in the previous 2 s and which are clearly longer for bacteria moving to the right. A net drift speed of $\bar{v}^* \sim 2$ μm s$^{-1}$ is obtained, and cells start to accumulate in the dark right margin of the observation area (See Supplementary Movie 1).

**Controllable velocity field.** We next study how to optimize cell rectification by tuning the feedback parameters. The radius of the spotlight $R$ must be clearly larger that the cell size to ensure a stable illumination against cell fluctuations in orientation and position. At the same time, $R$ should be small enough to reduce unwanted overlaps with spotlights from different cells. In all experiments we choose $R$ values in the range 2–4 μm. Our *E. coli* cells reorient by "tumbling", random reorientation events caused by transient unbundling of flagella[31] and occurring with rate $\lambda \approx 1$ s$^{-1}$. For this reason the delay time $\tau$ is naturally constrained to be significantly smaller than the persistence time of the

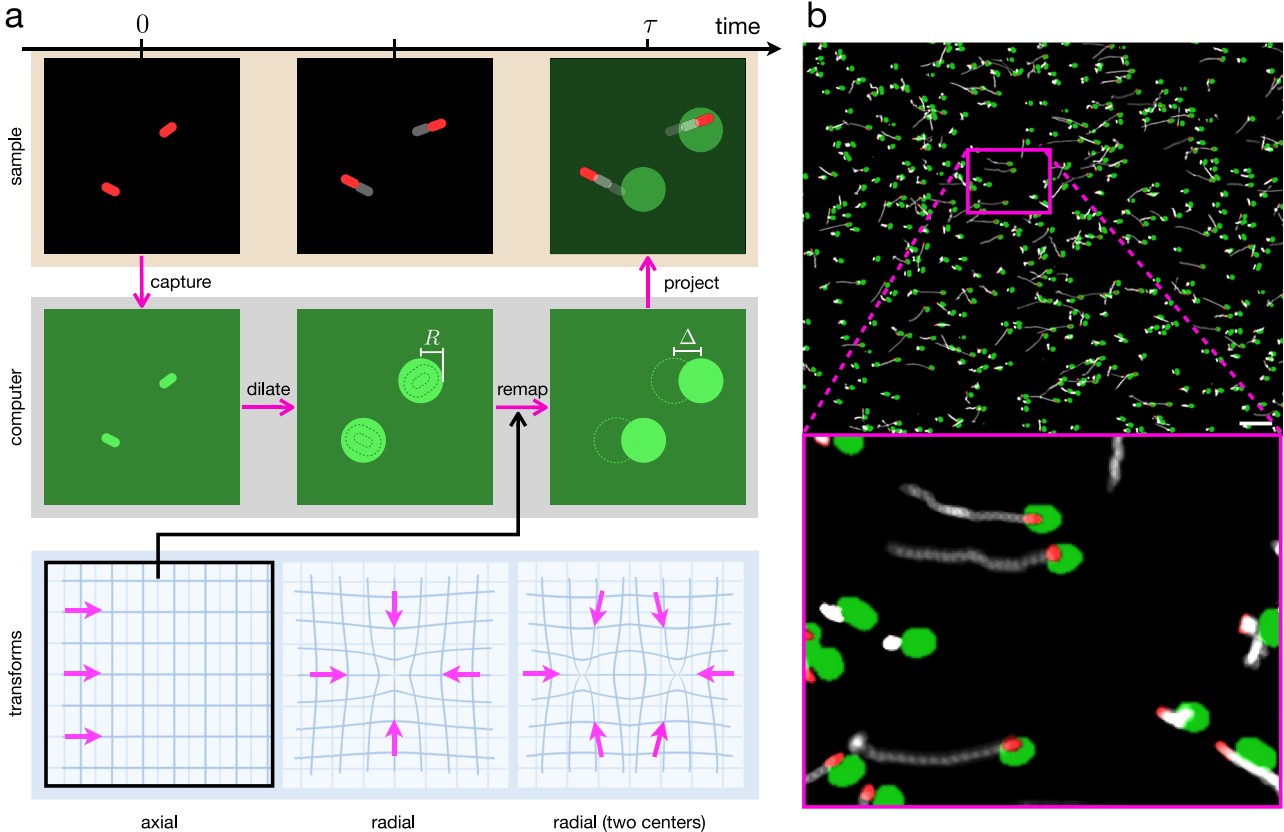

**Fig. 1 Optical feedback loop. a** Schematic representation of the feedback loop and image transformations. Red spots represent bacteria while light intensities are shown in green. First we capture a system's snapshot in dark field mode. Secondly we dilate the image using a circular kernel so that every cell is replaced by an approximately circular object of radius $R$. Then we remap this image by a plane transformation that moves the content of every pixel by a distance $\Delta$ towards the desired flow direction $\hat{n}$. In our experiments we considered axial and radial (one or two centers) transformations. Finally we project the transformed image back onto the system after a delay time $\tau$. **b** Experiment in which cells are guided towards the right. Dark-field microscope image of the bacteria (red) superimposed with the bright ($l_1$) regions in projected light pattern (green) with $R = 3.8\,\mu m$, $\Delta = 5.1\,\mu m$ and $\tau = 0.2\,s$. Bottom image is an enlargement of the box shown in the top. White traces represent bacterial trajectories and are obtained by superimposing images acquired in the previous 2 s. Scale bar is 40 μm. See Supplementary Movie 1.

trajectories (i.e. $\tau \ll \lambda^{-1}$) so we can assume that bacteria move in straight lines during the time interval $\tau$. We empirically find that delays in the range $0.1 - 0.5\,s$ produce optimal results (see Supplementary Fig. 1) and always keep $\tau \leq 0.2\,s$ in all experiments. To maximize net speed we want to be as close as possible to an angular speed dependence as that of Eq. (1). Ideally, for point-like bacteria, we could make sure that the speed is maximum for swimming directions $|\theta| < \pi/2$ when $\Delta$ is chosen as (see Fig. 2a):

$$\Delta^* = \sqrt{R^2 - (v_1 \tau)^2} \qquad (3)$$

For a given choice of $R$ and $\tau$, we experimentally scan different $\Delta$ values, track swimming cells and compute the drift speed $\bar{v}$ as the average $x$ component of cell velocities over time (Fig. 2b). Theoretically, for point-like cells, all swimming with the same speeds $v_0$ and $v_1$ when exposed to the two light levels $I_0$ and $I_1$, we would only expect to see a non-zero average speed in the interval $v_1 \tau - R < \Delta < v_1 \tau + R$ with a maximum speed $\bar{v}^*$ at $\Delta^*$. We measure a drift speed around $2.5\,\mu m\,s^{-1}$ for $\Delta V = v_1 - v_0 = 6.5\,\mu m\,s^{-1}$, which is in substantial agreement with the theoretical limit of $\bar{v}^* = 2.1\,\mu m\,s^{-1}$. The experimental results present a peak at a position that is close to $\Delta^*$ but much broader than expected. This broadening is possibly due to the finite cell size along with the large variability of swimming speeds in the sample[9,32]. In particular, for

the two mean speeds $v_1 = 10\,\mu m\,s^{-1}$ and $v_0 = 3.5\,\mu m\,s^{-1}$, the corresponding standard deviations are $\sigma_{v_1} = 3.5\,\mu m\,s^{-1}$ and $\sigma_{v_0} = 1.5\,\mu m\,s^{-1}$.

The polar plot of the instantaneous velocities (Fig. 2c, d) clearly shows that this method can produce strongly angular dependent speeds. For large displacements $\Delta \gg R$, all the bacteria are illuminated with low light intensity, and thus have isotropically minimum velocity modulus ($v_0$). On the other hand, for very small displacements $\Delta = 0$, the cells are always illuminated with a high intensity, thus the modulus of the velocity is maximum ($v_1$) for every direction $\theta$. The optimum case where the flow is maximized is of the order of $\Delta \sim R$, and the velocity field has maximum modulus for small $\theta$ and minimum modulus for large $\theta$. The experimental shape is similar to the theoretical Eq. (1) (discontinuous gray line in Fig. 2d), with a smoother transition around $|\theta| = \pi/2$.

**Optical confinement in high activity regions.** A common problem when dealing with light-driven particles is that we can usually modulate light over a finite field of view within an otherwise dark and non motile sample. In these situations we have a constant outflow of active particles that, once they enter the surrounding darkness, stop swimming and are lost forever. We now show that, using our dynamic feedback, we can use light to confine bacteria in regions of tunable size, density, and motility. The obtained steady states have a density that decays

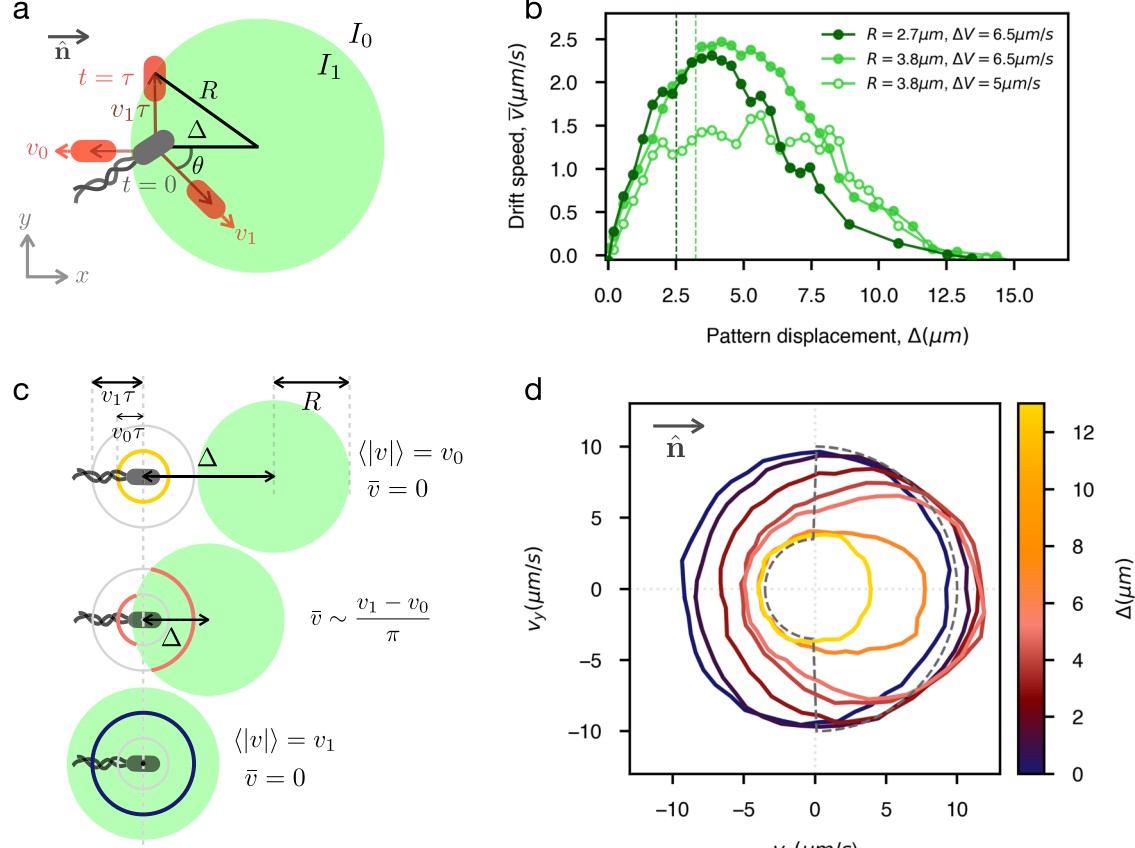

**Fig. 2 Rectification. a** Schematics of an optimal choice for feedback parameters resulting in maximum rectification towards the right. The white background corresponds to minimum light intensity $I_0$ with associated cell velocity modulus $v_0$ and the green areas to maximum light intensity $I_1$ and $v_1$. A circular spotlight of radius $R$ is shifted by a distance $\Delta = \Delta^* = \sqrt{R^2 - (v_1\tau)^2}$ (Eq. (3)) to the right of the cell's position at time $t = 0$. In red the different possible positions at time $t = \tau$ are represented showing that all cells swimming at an angle $|\theta| < \pi/2$ will move with maximum speed $v_1$. **b** Drift speed of bacteria versus the displacement $\Delta$ for different radii $R$ and velocities $v_1$, $v_0$. In all cases $\tau = R/2v_1$. The dashed lines are the corresponding theoretical optima according to Eq. (3). **c** Evolution of the polar speed plot as a function of $\Delta$. When $\Delta > R + v_1\tau$ cells will always be outside of their spotlight and swim with an isotropic speed $v_0$ (yellow). For $\Delta < R - v_0\tau$ cells will always be inside and swim with isotropic speed $v_1$ (blue). For intermediate values of $\Delta$ cells will have an angle dependent speed (red) swimming faster when moving towards right. **d** Polar speed plot for different values of $\Delta$ when $\hat{\mathbf{n}} = \hat{\mathbf{x}}$, $R = 2.7$ μm, $\tau = 0.1$ s, $v_1 = 10$ μm s$^{-1}$ and $v_0 = 3.5$ μm s$^{-1}$ (corresponding to the dark green line in **b**). Dashed line is the ideal speed profile for maximum flow (Eq. (1)).

exponentially towards the boundary, thus minimizing particle loss. Let us first theoretically analyze the structure of the steady state that is obtained if bacteria swim with the maximum speed $v_1$ in a central band of width $2a$. Outside of the band the speed has an angular dependence as in Eq. (1) where $\hat{\mathbf{n}} = -\hat{\mathbf{x}}$ for $x > a$ and $\hat{\mathbf{n}} = \hat{\mathbf{x}}$ for $x < -a$ (see Fig. 3a). For $x > a$, the number density $P(x, \theta, t)$ of bacteria swimming at position $x$ with an angle $\theta$ will evolve in time with the equation[33]:

$$\dot{P}(x, \theta, t) = -\partial_x P(x, \theta, t)v(\theta)\cos\theta - \lambda P(x, \theta, t) + \frac{\lambda\rho(x, t)}{2\pi} \quad (4)$$

where we assumed 2D run and tumble dynamics with a tumbling rate $\lambda$ and introduced the total number density $\rho(x, t) = \int P(x, \theta, t) d\theta$. The first term on the right side of Eq. (4) represents the divergence of the particles' current, the second one particle loss due to tumbling and reorientation, while the last term corresponds to the particle gain due to tumbling from all other directions into $\theta$. We now look for stationary states where the spatial and angular densities factorize as $P(x, \theta) = \rho(x)\chi(\theta)$:

$$-\partial_x \rho(x)\chi(\theta)v(\theta)\cos\theta - \lambda\rho(x)\chi(\theta) + \frac{\lambda\rho(x)}{2\pi} = 0 \quad (5)$$

which by separating variables becomes:

$$-\partial_x \log\rho(x) = \frac{\lambda}{v(\theta)\cos\theta}\left(1 - \frac{1}{2\pi\chi(\theta)}\right) = k \quad (6)$$

where $k$ is a constant whose value can be find by imposing the normalization condition:

$$1 = \int_0^{2\pi} \chi(\theta)d\theta = \int_0^{2\pi} \frac{1}{2\pi}\frac{1}{1 - kv(\theta)\cos\theta/\lambda}d\theta \quad (7)$$

A trivial solution is given by $k = 0$ and thus $\rho(x) = $ constant and $\chi = 1/2\pi$ corresponding to a steady homogeneous flow with mean speed $v^*$ and extending to infinity. To find a zero flux steady state we need a non vanishing solution for Eq. (7) which can be obtained by numerical solution or, with excellent approximation, by the analytic expression (see Supplementary Note 1):

$$k = \frac{2\lambda}{\pi}\frac{v_1 - v_0}{v_1 v_0} \quad (8)$$

Remarkably, for $x > a$ the corresponding density $\rho(x) \propto e^{-kx}$ decays exponentially on a length scale that, for typical speeds and tumbling rates, can be a few tens of micrometers. For $x < a$

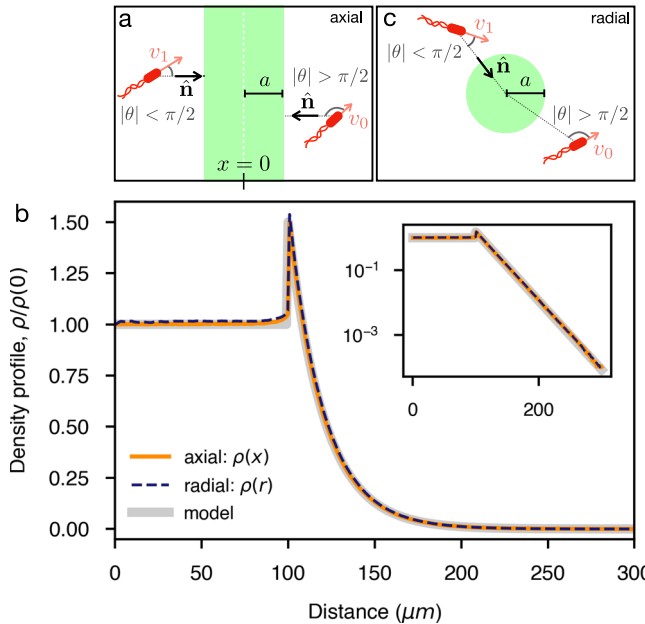

**Fig. 3 Run and tumble particles in 2D with axial and radial feedback.**
**a** Schematic showing the axial geometry to accumulate cells in a band of thickness $a$. **b** Simulation results for the stationary density profiles in cases of radial and axial rectification, and comparison with the theoretical model. Inset shows results in a semi-log plot. The parameters are $v_0 = 5\,\mu m\,s^{-1}$, $v_1 = 10\,\mu m\,s^{-1}$, $a = 100\,\mu m$ and $\lambda = 0.75\,s^{-1}$. Box size is $L = 750\,\mu m$. **c** Schematic showing the radial geometry.

the stationary density will be uniform while at the boundary $x = a$ the internal and external currents must satisfy the continuity conditions:

$$P(a^-, \theta)v_1 = P(a^+, \theta)v_1 \,, \quad |\theta| < \pi/2$$
$$P(a^-, \theta)v_1 = P(a^+, \theta)v_0 \,, \quad |\theta| > \pi/2$$

so that:

$$\rho(a^+) = \int_0^{2\pi} P(a^+, \theta)d\theta = \int_{|\theta| < \frac{\pi}{2}} P(a^+, \theta)d\theta + \int_{|\theta| > \frac{\pi}{2}} P(a^+, \theta)d\theta$$
$$= \int_{|\theta| < \frac{\pi}{2}} P(a^-, \theta)d\theta + \frac{v_1}{v_0}\int_{|\theta| > \frac{\pi}{2}} P(a^-, \theta)d\theta$$
$$= \left(1 + \frac{v_1}{v_0}\right)\frac{\rho(a^-)}{2}$$

$$(9)$$

And the full solution reads:

$$\rho(x) = \begin{cases} 1 & x < a \\ \frac{1}{2}\left(1 + \frac{v_1}{v_0}\right)e^{-k(x-a)} & x > a \end{cases} \quad (10)$$

To check this theoretical prediction we simulate non-interacting run and tumble particles with an angular dependent speed as in Fig. 3a when $|x| > a$, while moving at maximum speed $v_1$ for $|x| < a$ (see Fig. 3a). Simulation results are reported Fig. 3b showing an excellent agreement with Eq. (10). In an experimental system, however, this axial geometry would be still affected by cell leakage at the top and bottom boundaries. This could be solved by driving bacteria radially towards a circular region contained in the illuminated area (see Fig. 3c). Remarkably the solution in Eq. (10) still provides a near-perfect description of the density profile from simulations performed in the radial geometry, as expected when $a$ is much larger than the persistence length $\approx v_1/\lambda$. To

demonstrate this possibility experimentally, we modify the last step in the feedback algorithm by substituting the lateral shift in $\hat{\mathbf{n}} = \hat{\mathbf{x}}$ of the dilated image with a radial contraction by a constant displacement in the direction $\hat{\mathbf{n}} = -\hat{\mathbf{r}}$ (Fig. 3b). With this, we guide the bacteria towards a central circle of radius $a$ where we set a uniform light intensity $I_1$ corresponding to isotropic swimming with maximum speed $v_1$. Outside of the disk we get, for the optimal case $\Delta = \Delta^*$, a mean cell velocity that can be still approximated by Eq. (1), where now $\theta$ is the angle formed by the swimming direction and the inward radial vector $-\hat{\mathbf{r}}$. Starting from a uniform density at $t = 0$ min we turn on the radial feedback and observe a radial flow of bacteria moving towards the central circle shown in green (Fig. 4a), where the number of bacteria increases until a stationary density profile is reached in a few minutes (Fig. 4c). The timescale $\tau_r$ for this relaxation can be estimated as the time needed for bacteria to travel at the drift speed $\bar{v} \approx 2\,\mu m\,s^{-1}$ over a distance $L \approx 400\,\mu m$ from the edge to the center of the illuminated area. This gives $\tau_r \approx L/\bar{v} \approx 3$ min which is in agreement with experimental observations. With an initial radius of $a = 145\,\mu m$ we observe a 3 fold increase in bacterial density within the accumulation circle (Fig. 4c). The density profile of the stationary state is characterized by an exponential decay for $r > a$ with a fitted decay length of $\ell = 1/k = 19\,\mu m$. For a radius $a$ that is much larger than bacteria persistence length we expect, as also confirmed by simulations in Fig. 3b, that an exponential decay with the same $k$ as in Eq. (8) could still describe the steady state in the radial case. The fitted $k$ value is consistent with the theoretical prediction in Eq. (8) if we assume a tumbling rate $\lambda = 0.8\,s^{-1}$ together with the measured values for $v_0 = 5\,\mu m\,s^{-1}$ and $v_1 = 10\,\mu m\,s^{-1}$. The quantitative agreement between experiments and the theoretical predictions for the non-interacting run and tumble model, also suggests that interactions, steric or hydrodynamic, do not play a relevant role here. When we keep decreasing $a$, we find that internal density increases up to ~7 fold with respect to the initial uniform density, while the total number of cells is conserved. The relaxation time is much shorter in this case, due to the fact that the travel distance is just the change in radius $\Delta a = 18\,\mu m$ instead of $L$. We keep each radius fixed for 10 min to get a well averaged density profile and verify the stability of the method over long timescales. This proves that our method can indeed be used to optically concentrate and confine thousands of actively swimming bacteria within regions of independently tunable density and activity (Fig. 4b, c). All stationary density profiles in Fig. 4d can be fitted by Eq. (10) with $k$ and a global factor as the free fit parameters. The agreement of the fits improves for larger confinement radius $a$ which can be partly due to the fact that we assume $a$ much larger than persistence length in the theoretical model. Ultimately, this is a fully reversible process, i.e. when switching off the feedback loop and applying constant illumination to the full sample area, the accumulation dissolves in just a few seconds (Fig. 4d). This provides further evidence of the high activity of the confined cells (see Supplementary Movie 3).

As a further feature of our method we show that it is possible to split, relocate and merge again optically confined clouds of highly motile bacteria (Fig. 5, see Supplementary Movie 4). To do that we apply our optical feedback with a radial transform that constantly drives bacteria towards two centers (Fig. 1a). This allows to effectively redistribute bacteria in space conserving the total number of trapped cells as shown in the density profiles in Fig. 5. This is not the case when using a sequence of static patterns[9] for dynamic density shaping of photokinetic bacteria: while we could concentrate bacteria inside a dark disk within an illuminated area and then move the disk to transport bacteria, by moving the dark disc the bacteria on the trailing edge will be re-exposed to light and, for the most part, swim away forever.

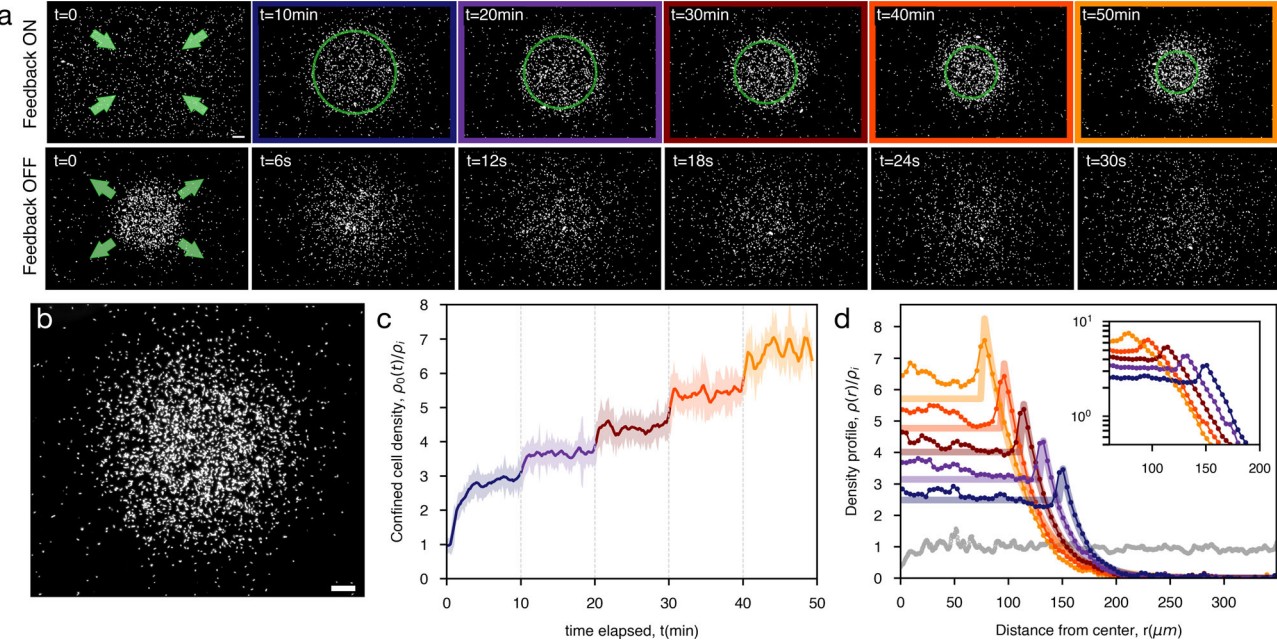

**Fig. 4 Optical confinement of highly motile bacteria.** Using the optical feedback loop with a radial transformation we confine bacteria in circular regions of radii $a$ that are progressively shrunk every 10 min. Different colors correspond to different $a = 145, 127, 109, 91, 73 \, \mu m$ At the end of the experiment, the feedback loop is turned off. Parameters are $\Delta = 1.8 \, \mu m$, $R = 2 \, \mu m$, $\tau = 0.1 \, s$, $v_1 = 10 \, \mu m \, s^{-1}$ and $v_0 = 5 \, \mu m \, s^{-1}$. See Supplementary Movie 2. **a** Assembly and disassembly of confinement. Dark-field microscope images at different times of bacteria accumulating towards the central circle of different $a$. Scale bar is 100 $\mu m$. **b** Steady state of confined bacteria. Feedback loop parameters are $\Delta = 3.3 \, \mu m$, $R = 2.7 \, \mu m$, $\tau = 0.1 \, s$, $a = 91 \, \mu m$, $v_1 = 10 \, \mu m \, s^{-1}$ and $v_0 = 5 \, \mu m \, s^{-1}$. Scale bar is 40 $\mu m$. See Supplementary Movie 3. **c** Corresponding evolution along time of the density inside the circle $\rho_0$, for $r < a$. Dotted gray lines indicate when $a$ changes. **d** Stationary state profiles for the different $a$, each averaged over 5 min. Gray points are the initial state. Thick lines are fits to the theoretical Eq. (10), with an average resulting decay length of $1/k = 24 \pm 2 \, \mu m$. Error bars (standard error of the mean) are smaller than the symbol size. Inset shows the data in a semi-log plot.

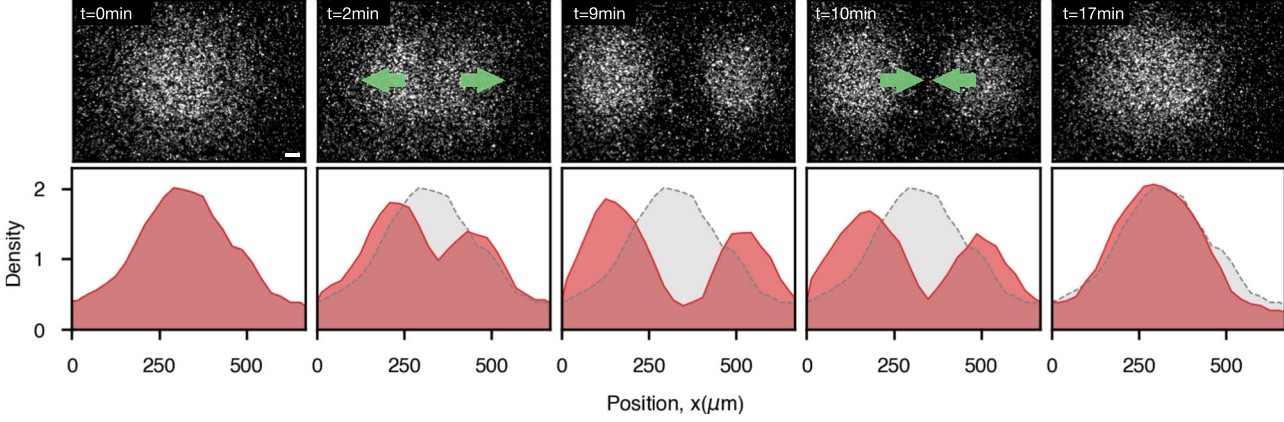

**Fig. 5 Splitting and merging of optically confined clouds of motile bacteria.** Using a radial transform with two centers we can split an optically confined region of highly motile bacteria into two separate clouds and then merge them together, always conserving the total number of trapped cells. Bottom panel shows the corresponding density profiles (red) overlapped to the initial profile at $t = 0$ (gray). Scale bar is 100 $\mu m$. See Supplementary Movie 4.

## Discussion

In summary, we demonstrated an optical feedback loop that couples swimming bacteria to their past configurations, thereby breaking time reversal symmetry and rectifying their random walks into designable flow fields. The feedback is based on simple geometric transformations of a system image taken at an earlier time, so it is robust and scalable to large and concentrated suspensions, although a decrease in efficiency is expected when interparticle distances become comparable with the spotlight size. We show that bacteria swimming in this dynamic structured illumination can be confined in regions with simultaneous high density and high activity. As predicted by a run and tumble

dynamics with anisotropic speed, the stationary density of these optically confined states is characterized by rapidly decaying exponential tails that prevent cell escape and allow the total number of particles to be conserved for hours, as opposed to other approaches that accumulate slow bacteria and deplete density over time[8,9]. By dynamically changing the applied geometric transformation we can split, relocate and finally merge these optically confined clouds of swimming bacteria always preserving the total cell number. Although we used photokinetic *E. coli* the method could be easily adapted to many natural or synthetic microswimmers with light controlled motility. This research also opens up many new directions for future

developments, including the possibility of having adjacent bacterial clouds with independently controlled density and motility parameters to study non-equilibrium phenomena such as active pressure. Additionally, the use of more complex convolution kernels for dilation could, for example, help sort bacteria by their maximum speed. More generally, optical transport and confinement of active matter will provide a versatile tool to produce reconfigurable active baths for controllable powering of micro-machines or systematic studies of non-equilibrium phenomena in active systems.

## Methods

**Bacterial Strain**. For all the experiments we used the *E. coli* strain AUG, constructed using the *E. coli* AB1157 obtained from the CGSC (Coli Genetic Stock Center). Using lambda red recombination with the recombination plasmid pKD46[34] we deleted the operon enconding the ATP Synthase complex as in ref. [8] (corresponding to the position 3,915,643-3,922,426 of the *E. coli* MG16555 chromosome). We replaced it with the CmR cassette flanked by 50 bp homology amplified using as template pKD3 (see Supplementary Note 2). PCR reactions confirmed the desired genotype, we thus obtained a strain AB1557 Δ *unc::cmR*. Then we transformed it with the plasmid pBAD-His C encoding the SAR86 γ-proteobacterial photorhodopsin (PR) (a kind gift from Judith Armitage, University of Oxford)[35].

**Sample preparation**. *E. coli* colonies from frozen stocks are grown overnight at 30 °C on LB agar plates supplemented with ampicillin (100 μg/ml). A colony is picked and cultivated overnight at 30 °C at 200 rpm in 10 ml of LB with ampicillin. The next day, the overnight culture is diluted 100 times into 5 ml of TB containing ampicillin, and grown at 30 °C, 200 rpm for 4h. Then all-*trans*-*retinal* (20 μM) and L-arabinose (1 mM) are added to ensure expression and proper folding of PR in the membrane. The cells are collected after 1 hour of induction by centrifugation at 1300 × *g* for 5 min at room temperature. The supernatant is removed and the bacterial pellet is resuspended in a motility buffer, containing 10mM potassium phosphate (pH 7.0), 0.1 mM EDTA (pH 7.0), 0.02% Tween20 and L-Lactate 10 mM. This washing procedure is repeated three times in total, then the cells are resuspended at the desired concentration (OD). This medium allowed the cells to be motile without allowing growth or replication, so the concentration of the cells remained constant throughout the experiments.

2 μL of the prepared bacterial suspension is injected into a glass chamber (Leja, 10 μm sample), hermetically sealed at both sides with UV glue (Norland Optical Adhesive NOA81). Oxygen is depleted by bacteria in around 30–45 min for OD = 0.5. Once this has happened, the bacteria swim only where there is green light, and their speed increases with the intensity of the light.

**Microscopy and light projection**. We use a custom inverted optical microscope in dark field mode, equipped with a 10x magnification objective (Nikon; NA=0.30) and a high-sensitivity CMOS camera (Hamamatsu Orca-Flash 2.8). Light patterns are generated using a digital light processing (DLP) projector (Texas Instruments DLP Lightcrafter 4500) coupled to the same microscope objective used for imaging. Green light from the DLP projector is filtered by a bandpass filter (central wavelength 520 nm) and then coupled to the microscope objective through a dichroic mirror. A long pass filter prevents illumination light to reach the camera, as in ref. [9]. The size of a DLP projector pixel imaged on the sample plane is 0.75 μm.

## Data availability

The data that support the findings of this study are available from the corresponding author upon request.

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

## Acknowledgements

The research leading to these results has received funding from the European Research Council under the ERC Grant Agreement No. 834615 (R.D.L.).

## Author contributions

H.M.-C., C.M., G.F. and R.D.L. designed the experiments. H.M.-C. performed experiments and analyzed data. C.M. performed simulations. G.F. was responsible for the strain

construction. G.F. and H.M.-C. were responsible for the growth of bacterial strains. H.M.-C., C.M., G.F. and R.D.L. wrote the manuscript.

## Competing interests

The authors declare no competing interests.
