## [Peer review file · Nature Communications]

REVIEWER COMMENTS

Reviewer #1 (Remarks to the Author):

This paper puts forward an intriguing idea of steering bacteria with optical programmed stimuli which are implemented in such a way such the optimal signal is stemming from the past history of the bacteria. A proof-of-principle experiment is designed and described to confine and rectify bacterial flow. In general this is an important idea which could be published in Nature Communications. But the manuscript lacks clarity in certain aspects.

The authors should carefully and significantly improve the manuscript including the following directions:

1) The abstract is difficult to read, same applies for the introduction. I got the real idea only when Figure 1 was discussed.

``what does they swim under a projected image of their past'', what is the ``geometric transformation that is applied'', these are obscure expressions. ``The obtained currents achieve a maximum value'': maximum as a function of what? ``Steep exponential tails'' are mentioned but it is not said as a function of what (I conjectured time here but later I learnt it was space).

Same applies for the paragraph in the introduction which starts with ``Here we demonstrate'' This should be extended and much better explained.

2) page 1 ``This is because in steady states the product ρv must be constant'' This requires many approximations which is possibly oversimplified

in the present context.

3) Figure 3d shows density profiles, the inset on a semi-logarithmic plot. The authors claim an exponential decrease. First of all this

conclusion is based on a linear behavior over an incomplete decade only. What are the error bars?

4) In the SI the authors propose a simple run-and-tumble model

to backup the experimental data with a model calculation. What is the role of steric and hydrodynamic

interactions between the bacteria?

It should not be very difficult

in principle to incorporate those into the simulation model on a pairwise level.

5) Figure 2b: here the drift speed of bacteria is shown. A precise definition of the drift speed should be given.

6) Figure 4: The splitting and merging seems to be a bit obvious. Can the relative number contained in the two clouds be steered and controlled?

7) page 1: It is not easy to see the connection between the Maxwell daemon and the message of the paper.

8) The following relevant literature should be cited and discussed briefly:

C. Lozano, and C. Bechinger, Nature Communications 2495 (2019);

H. Merlitz, H. D. Vuijk, J. Brader, A. Sharma, and J.-U. Sommer,

The Journal of Chemical Physics 148, 194116 (2018).

Minor:

9) typos: ``sistematic" on page 6, ``ca", ``theta", ``wit" on page 2 of the SI

10) References [16], [23], [24]. [25], [26] contain strange initials.

Reviewer #2 (Remarks to the Author):

The manuscript by Massana-Cid et al. introduces a novel optical feedback method for rectifying swimming *E. coli* bacteria and accumulate them into high- and low-density regions. The experimental results are clearly presented and well verified by theoretical arguments that quantitatively connecting with the experimental results.

I have no major scientific objections against neither the results or their presentation. The key point is instead whether the novelty of this method is large enough compared to the previously published one by the same group (Ref. [9]). The method is distinctly different in that it uses feedback to steer bacteria in a predefined direction (creating a net flux), while the previous one used optical feedback applied to the density field to accumulate bacteria into dense and dilute regions of a prespecified shape. The current method is more versatile in that it doesn't require high-density regions to have low motility, and can therefore also be used to prevent particle escape and absorption into low-motility regions, as demonstrated in the second part of the article. All in all, I think these differences add enough novelty to merit publication in Nature Communications, although I remain somewhat undecided.

Some minor questions and comments:

1. The sentence "This is because in steady states the product ρv must be constant throughout the sample, if it is set to zero in the surrounding dark area where the particle velocity vanishes, then it must be zero everywhere so regions of nonvanishing density can only be composed of non-motile bacteria" could be written in a better way. While correct in a mathematical sense, in real systems there will always be translational diffusion, and ρv will furthermore be non-constant due to lag time in the response to the light field etc. I'd rather suggest writing something like "...because the product ρv is approximately constant throughout the system, bacteria will over time be absorbed by the low-motility surroundings".

2. "Demon" is misspelled throughout (unless Maxwell referred to the computer science term).

3. Regarding the constraint $\tau < \lambda^{-1}$: Presumably, the tumble rate is not constant but Poisson distributed, meaning that there will be many tumble events shorter than the average λ taking place. Would this not put a stricter constraint on the choice of τ ?

4. The light (yellow, orange) lines in Fig. 3 are difficult to read in print.

5. The presentation of the theoretical analysis can be improved for clarity. First, it is not immediately clear what the geometrical setting is, as there is a sudden jump between Cartesian and polar coordinates. I'd suggest starting with a small figure showing the coordinate system and geometry with the two velocity functions (rather than saying "we set $\hat{n} = -\hat{x}$ "). After (10), there is then a sudden jump to "...actively move bacteria in a negative radial direction...", which is not clear, as the analysis is done in a one-dimensional Cartesian coordinate system. Furthermore, the second time " $\hat{n} = \hat{x}$ " occurs in scalar notation - although I think this should anyway be written in a clearer way.

6. Regarding the numerical simulations, I think these should not be relegated to the SI, but rather included as a separate panel in one of the existing figures.

7. Why is the decrease in R_c so slow (every ten minutes)? Wouldn't the theoretical timescale of the rectification process be comparable to the tumbling time, which is seconds rather than minutes? Could the authors comment on this in the paper, including whether there is any theoretical justification for this?

Reviewer #3 (Remarks to the Author):

The authors of this manuscript report beautiful experiments on the light-controlled dynamics of bacterial systems. In particular, they develop a robust control scheme that identifies bacteria and projects light patterns to guide their motion. Crucially, the scheme does not require tracking but relies on creating illumination masks based on dark-field images of the bacteria.

The authors robustly show that their scheme can rectify the swimming of the bacteria and thus can be used to provide robust means to collectively guide and accumulate them. They also show simple theoretical arguments to predict the density distributions under different conditions and compare the results with numerical simulations.

This work belongs to a very exciting field of research that deals with external control of active systems and constitutes a significant advance, which can be of inspiration for others. The application of the rectification scheme is very clever and the consequences remarkable.

I therefore strongly recommend publication. I however have some detailed questions that I would like to ask the authors to consider before publication.

- The rectification scheme appears to work for most cells, but in the first supporting movie that are clear cases of bacteria that seem to be constantly missing the bright spot but nonetheless move with high speed at angles close to 90 deg. Can the authors comment on these cases?
- Can the authors comment on the possibility of using another shape of the illumination mask to direct the bacteria even more efficiently? Would for instance be beneficial to have a wedge-shaped mask to change the angular dependence of the velocity in equation 1?
- Can the authors describe more precisely the meaning to of the second and third term in the right hand-side of equation 4?
- In the data of figure 3 there is a clear peak in density at the edge of the pattern, in proximity, but not quite at, a position corresponding to R_c . Can the authors comment on the physical origin of this feature? Is this related to the persistence length/tumbling rate in relation to the value of τ chosen for the control? Is this accumulation at the edge a consequence of a finite delay in adjusting the velocity for outgoing cells? In fact, this peak does not seem to be present in the simulated data with instantaneous feedback shown in the SI. Am I right to think that this peak should be at a position relative to R_c of the order of the distance traveled over τ ?

NCOMMS-22-02620A

“Rectification and confinement of photokinetic bacteria in an optical feedback loop”

by H. Massana-Cid, C. Maggi, G. Frangipane and R. Di Leonardo

Authors’ reply to Reviewers comments

We thank all the referees for their thorough reading of our manuscript and for the constructive suggestions and criticisms. Our point-by-point responses and the corresponding changes to the manuscript are described below.

Response to Reviewer #1

Reviewer: *This paper puts forward an intriguing idea of steering bacteria with optical programmed stimuli which are implemented in such a way such the optimal signal is stemming from the past history of the bacteria. A proof-of-principle experiment is designed and described to confine and rectify bacterial flow. In general this is an important idea which could be published in Nature Communications. But the manuscript lacks clarity in certain aspects.*

Authors: We thank the Reviewer for the positive comments on our manuscript. We have considered all the recommendations and improved the manuscript accordingly.

Reviewer: *The authors should carefully and significantly improve the manuscript including the following directions:*

1) *The abstract is difficult to read, same applies for the introduction. I got the real idea only when Figure 1 was discussed.*

“what does they swim under a projected image of their past”, what is the “geometric transformation that is applied”, these are obscure expressions. “The obtained currents achieve a maximum value”: maximum as a function of what? “Steep exponential tails” are mentioned but it is not said as a function of what (I conjectured time here but later I learnt it was space).

Authors:

We introduced new text, explanations and figures and we believe that the revised manuscript has significantly gained in clarity.

Following the Reviewer’s suggestions we changed the abstract as follows: (changes marked in blue)

“Active particles can self-propel by exploiting locally available energy resources. When powered by light, these resources can be distributed with a high resolution allowing spatio-temporal modulation of motility. Here we show that the random walks of light-driven bacteria are rectified when they swim in a structured light field that is obtained by a simple geometric transformation of a previous system snapshot. The obtained currents achieve an optimal value that we establish by general theoretical arguments. This optical feedback is used to gather and confine bacteria in high-density and high-activity regions that can be dynamically relocated and reconfigured. Moving away from the boundaries of these optically

confined states, the density decays to zero in a few tens of micrometers, exhibiting steep exponential tails that suppress cell escape and ensure long-term stability. Our method is general and scalable, providing a versatile tool to produce localized and tunable active baths for microengineering applications and systematic studies of non-equilibrium phenomena in active systems.”

Reviewer: *Same applies for the paragraph in the introduction which starts with “Here we demonstrate” This should be extended and much better explained.*

Authors: In the main text, we add the the following text to improve in clarity (added sentence in blue):

“Here we demonstrate that, by projecting back onto a suspension of photokinetic bacteria a properly transformed image of the system at an earlier time, we can guide cells with maximum efficiency along a designable flow field. Specifically, an optical feedback keeps the cells under a binary illumination pattern such that only bacteria moving in the desired direction are illuminated by a bright green spotlight and swim fast. Other cells are kept under a background of low light intensity and swim slower until a tumble turns them in the right direction. We exploit this method to optically gather and confine large collections of cells in high density and high activity regions over long time periods.”

Reviewer: *2) page 1 “This is because in steady states the product ρv must be constant” This requires many approximations which is possibly oversimplified in the present context.*

Authors: We agree with the reviewer that this is an approximation, so we write instead

“This is because structured illumination is confined by the finite size of the spatial light modulator, so that the system is surrounded by a dark region of particles with $v=0$. Since the product ρv must remain approximately constant throughout the system, bacteria will over time be absorbed by the low-motility surroundings.”

Reviewer: *3) Figure 3d shows density profiles, the inset on a semi-logarithmic plot. The authors claim an exponential decrease. First of all this conclusion is based on a linear behavior over an incomplete decade only. What are the error bars?*

Authors: Errorbars (Standard Error of the Mean) are smaller than symbol size so we did not plot them. But this is important information that was missing in the previous version so now explicitly say it in the figure caption. Although, as the Reviewer correctly points out, a decade could be not enough to identify the exact decay law, a semilog scale provides a better linearization of the curves over the largest interval of distances, as shown by the comparison with a log-log scale plot in the figures below:

We would also like to stress that our theoretical model makes a precise prediction of the full density profile (including a discontinuity and exponential tails) that closely match all experimental densities as shown in Fig.4d and make us confident in its validity.

Reviewer: 4) *In the SI the authors propose a simple run-and-tumble model to backup the experimental data with a model calculation. What is the role of steric and hydrodynamic interactions between the bacteria? It should not be very difficult in principle to incorporate those into the simulation model on a pairwise level.*

Authors: Indeed, both steric and hydrodynamic interactions play a role in very dense systems. In our case, however, bacteria are not strongly confined in 2D (10 μm height chamber) so that reorientation through collisions is very rare. On the other hand, confinement by solid walls reduces hydrodynamic couplings. In the end, we believe that in our case, large density will affect the feedback mechanism much earlier than steric and hydrodynamic interactions become important. Therefore, while it would not be difficult to incorporate pairwise interactions into our simulations, we believe that these will not lead to better agreement with the experimental data or provide a better understanding of the physical nature of the observed phenomena. This is not true if bacteria were strictly confined in 2D and steric interactions would then become very strong. We are currently performing experiments in these conditions and interactions will be an important ingredient there. We now explicitly say it in the text:

“The quantitative agreement between experiments and the theoretical predictions for the non-interacting run and tumble model, also suggests that interactions, steric or hydrodynamic, do not play a relevant role here.”

Reviewer: 5) *Figure 2b: here the drift speed of bacteria is shown. A precise definition of the drift speed should be given.*

Authors: We add in the text the definition of the drift speed:

“For a given choice of R and τ , we experimentally scan different Δ values, track swimming cells and compute the drift speed \overline{v} as the average x component of cell velocities over time (Fig.2b).”

We also changed the y axis label to “drift speed” for consistency.

Reviewer: 6) Figure 4: The splitting and merging seems to be a bit obvious. Can the relative number contained in the two clouds be steered and controlled?

Authors: Symmetric splitting and merging was meant to demonstrate that our bacterial clouds can be reconfigured and moved in space, conserving the total number of cells. We are currently exploring the possibility of using optical feedback for more complex operations. The one suggested by the reviewer is quite interesting and we may be able to unbalance the density by moving the two centers at different speeds. However, we feel that a more systematic exploration of the possibilities of our technique would be better suited for a separate article. We have expanded the conclusions by adding this and other suggestions from reviewer 3 as interesting future perspectives on this work:

This research also opens up many new directions for future developments, including the possibility of having adjacent bacterial clouds with independently controlled density and motility parameters to study non-equilibrium phenomena such as active pressure. Additionally, the use of more complex convolution kernels for dilation could, for example, help sort bacteria by their maximum speed.

Reviewer: 7) page 1: It is not easy to see the connection between the Maxwell daemon and the message of the paper.

Authors: We now better explain the connection in the revised manuscript:

“Let us suppose that one has complete knowledge of the individual positions and velocities of active particles. Such a “Maxwell demon”³⁰ could use this information to generate a net flow in a desired direction $\hat{\mathbf{n}}$ by adjusting individual particles' speeds.”

And later on:

“Like a “Maxwell demon”, the feedback loop rectifies the motion of bacteria by “opening or closing the door”, i.e. bright or dark exposure, depending on the direction of motion of the particles.”

Furthermore, we add a new citation for reference:

[30] Leff, H. & Rex, A. F. *Maxwell's Demon 2. Entropy, Classical and Quantum Information, Computing* (CRC Press, 2002)

Reviewer: 8) The following relevant literature should be cited and discussed briefly:

C. Lozano, and C. Bechinger, *Nature Communications* 2495 (2019);

H. Merlitz, H. D. Vuijk, J. Brader, A. Sharma, and J.-U. Sommer, *The Journal of Chemical Physics* 148, 194116 (2018).

Authors: We thank the reviewer for pointing us to these two relevant papers that are now included in the revised manuscript:

[12] C. Lozano, and C. Bechinger, *Nature Communications* 2495 (2019);

[15] H. Merlitz, H. D. Vuijk, J. Brader, A. Sharma, and J.-U. Sommer, *The Journal of Chemical Physics* 148, 194116 (2018).

Reviewer: Minor:

9) typos: “sistematic” on page 6, “ca”, “theta”, “wit” on page 2 of the SI

10) References [16], [23], [24]. [25], [26] contain strange initials.

We fix the mentioned typos on the main text and on the SI.

Response to Reviewer #2

Reviewer: *The manuscript by Massana-Cid et al. introduces a novel optical feedback method for rectifying swimming E. coli bacteria and accumulate them into high- and low-density regions. The experimental results are clearly presented and well verified by theoretical arguments that quantitatively connecting with the experimental results.*

I have no major scientific objections against neither the results or their presentation. The key point is instead whether the novelty of this method is large enough compared to the previously published one by the same group (Ref. [9]). The method is distinctly different in that it uses feedback to steer bacteria in a predefined direction (creating a net flux), while the previous one used optical feedback applied to the density field to accumulate bacteria into dense and dilute regions of a prespecified shape. The current method is more versatile in that it doesn't require high-density regions to have low motility, and can therefore also be used to prevent particle escape and absorption into low-motility regions, as demonstrated in the second part of the article. All in all, I think these differences add enough novelty to merit publication in Nature Communications, although I remain somewhat undecided.

Authors: We thank the reviewer for appreciating the novel aspects of our work and for raising a number of questions and comments that helped us improve the manuscript as detailed in the responses below:

Reviewer: *Some minor questions and comments:*

1. The sentence "This is because in steady states the product ρv must be constant throughout the sample, if it is set to zero in the surrounding dark area where the particle velocity vanishes, then it must be zero everywhere so regions of nonvanishing density can only be composed of non-motile bacteria" could be written in a better way. While correct in a mathematical sense, in real systems there will always be translational diffusion, and ρv will furthermore be non-constant due to lag time in the response to the light field etc. I'd rather suggest writing something like "...because the product ρv is approximately constant throughout the system, bacteria will over time be absorbed by the low-motility surroundings".

Authors: We agree with the reviewer that this is an approximation, so we write instead: "This is because structured illumination is confined by the finite size of the spatial light modulator, so that the system is surrounded by a dark region of particles with $v=0$. Since the product ρv must remain approximately constant throughout the system, bacteria will over time be absorbed by the low-motility surroundings."

Reviewer: *2. "Demon" is misspelled throughout (unless Maxwell referred to the computer science term).*

Authors: We thank the reviewer for making us discover that a daemon as in Linux is not the same as a demon as in Maxwell. We now spell it correctly throughout the manuscript.

Reviewer: 3. Regarding the constraint $\tau < \lambda^{-1}$: Presumably, the tumble rate is not constant but Poisson distributed, meaning that there will be many tumble events shorter than the average λ taking place. Would this not put a stricter constraint on the choice of τ ?

Authors: As the Reviewer correctly says, tumble events are Poisson distributed with a mean tumble rate λ so that the probability of having zero tumbles in a time interval τ is given by $e^{-\lambda\tau}$ so that, in principle τ should be significantly smaller than $1/\lambda$. This is now better discussed in the revised manuscript:

“For this reason the delay time τ is naturally constrained to be significantly smaller than the persistence time of the trajectories (i.e. $\tau \ll \lambda^{-1}$) so we can assume that bacteria move in straight lines during the time interval τ . We empirically find that delays in the range 0.1-0.5s produce optimal results (see Supplementary Information) and always keep $\tau \leq 0.2$ s in all experiments.”

Reviewer: 4. The light (yellow, orange) lines in Fig. 3 are difficult to read in print.

Authors: We thank the referee for noticing this and change the colormap so that it's easier to read in print:

Reviewer: 5. The presentation of the theoretical analysis can be improved for clarity. First, it is not immediately clear what the geometrical setting is, as there is a sudden jump between Cartesian and polar coordinates. I'd suggest starting with a small figure showing the coordinate system and geometry with the two velocity functions (rather than saying "we set $\hat{n} = -\hat{x}$ "). After (10), there is then a sudden jump to "...actively move bacteria in a negative radial direction...", which is not clear, as the analysis is done in a one-dimensional Cartesian coordinate system. Furthermore, the second time " $\hat{n} = \hat{x}$ " occurs in scalar notation - although I think this should anyway be written in a clearer way.

6. Regarding the numerical simulations, I think these should not be relegated to the SI, but rather included as a separate panel in one of the existing figures.

Authors: We thank the Reviewer for this suggestions that led us to expand the theoretical section with:

- Description and discussion of simulation results in axial and radial case

- Addition of a new Fig3 with simulation results, also featuring a diagram providing visual support for understanding the coordinate systems we used in the theoretical analysis.

We add:

“To check this theoretical prediction we simulate non-interacting run and tumble particles with an angular dependent speed as in Fig.3a when $|x| > a$, while moving at maximum speed v_1 for $|x| < a$ (see Fig.3a). Simulation results are reported Fig.3b showing an excellent agreement with Eq.(10). In an experimental system, however, this axial geometry would be still affected by cell leakage at the top and bottom boundaries. This could be solved by driving bacteria radially towards a circular region contained in the illuminated area (see Fig.3c). Remarkably the solution in Eq.(10) still provides a near-perfect description of the density profile from simulations performed in the radial geometry, as expected when a is much larger than the persistence length $\approx v_1/\lambda$.”

Reviewer: 7. Why is the decrease in R_c so slow (every ten minutes)? Wouldn't the theoretical timescale of the rectification process be comparable to the tumbling time, which is seconds rather than minutes? Could the authors comment on this in the paper, including whether there is any theoretical justification for this?

Authors: Indeed R_c (which we now renamed to “a” in the manuscript for simplicity and cohesiveness) could be decreased much faster, the experiment is just to show that the states are stable over long periods of time (overall experiment is almost 1h) and to ensure we are in the stationary state. Fig4c shows that the time needed to reach the stationary state starting from an homogeneous condition is of the order of a couple of minutes. Relaxation time is faster in subsequent steps in radius R_c . We agree with the Reviewer that this observation deserved to be discussed in the main text and add the following two paragraphs:

“The timescale τ_r for this relaxation can be estimated as the time needed for bacteria to travel at the drift speed $\overline{v} \approx 2 \mu\text{m/s}$ over a distance $L \approx 400 \mu\text{m}$ from the edge to the center of the illuminated area. This gives $\tau_r \approx L/\overline{v} \approx 3$ min which is in agreement with experimental observations. With an initial radius of $a=145 \mu\text{m}$ we observe a 3 fold increase in bacterial density within the accumulation circle (Fig.4c).”

...

“The relaxation time is much shorter in this case, due to the fact that the travel distance is just the change in radius $\Delta a=18 \mu\text{m}$ instead of L . We keep each radius fixed for 10 minutes to get a well averaged density profile and verify the stability of the method over long timescales.”

Response to Reviewer #3

Reviewer: The authors of this manuscript report beautiful experiments on the light-controlled dynamics of bacterial systems. In particular, they develop a robust control scheme that

identifies bacteria and projects light patterns to guide their motion. Crucially, the scheme does not require tracking but relies on creating illumination masks based on dark-field images of the bacteria. The authors robustly show that their scheme can rectify the swimming of the bacteria and thus can be used to provide robust means to collectively guide and accumulate them. They also show simple theoretical arguments to predict the density distributions under different conditions and compare the results with numerical simulations. This work belongs to a very exciting field of research that deals with external control of active systems and constitutes a significant advance, which can be of inspiration for others. The application of the rectification scheme is very clever and the consequences remarkable. I therefore strongly recommend publication. I however have some detailed questions that I would like to ask the authors to consider before publication.

Authors: We thank the referee for the very positive comments on our manuscript and for recommending publication. We respond to all of the raised questions below.

Reviewer: - *The rectification scheme appears to work for most cells, but in the first supporting movie that are clear cases of bacteria that seem to be constantly missing the bright spot but nonetheless move with high speed at angles close to 90 deg. Can the authors comment on these cases?*

Authors: When looking at the movie one should keep in mind that every single cell swims faster when moving to the right than if moving to the left and this appears clearly in those bacteria that change direction throughout the movie. At the same time, bacteria have a wide distribution of high (v_1) and low (v_0) velocities so that there can be cells that swim to the left with a comparable speed to that of another cell swimming towards the right.

Reviewer: - *Can the authors comment on the possibility of using another shape of the illumination mask to direct the bacteria even more efficiently? Would for instance be beneficial to have a wedge-shaped mask to change the angular dependence of the velocity in equation 1?*

Authors: That is a very interesting suggestion that could be easily implemented by convolving the binarized image with a kernel of arbitrary geometry. For example one could use tailored masks to sort particles according to their individual speeds. We have expanded the conclusions by adding this and other suggestions from reviewer 1 as interesting future perspectives on this work:

“This research also opens up many new directions for future developments, including the possibility of having adjacent bacterial clouds with independently controlled density and motility parameters to study non-equilibrium phenomena such as active pressure. Additionally, the use of more complex convolution kernels for dilation that could, for example, help sort bacteria by their maximum speed.”

Reviewer: - *Can the authors describe more precisely the meaning to of the second and third term in the right hand-side of equation 4?*

Authors: In the new version we now describe all terms in the right-hand side of the equation:

The first term on the right side of Eq. (4) represents the divergence of the particles' current, the second one particle loss due to tumbling and reorientation, while the last term corresponds to the particle gain due to tumbling from all other directions into θ .

Reviewer: - *In the data of figure 3 there is a clear peak in density at the edge of the pattern, in proximity, but not quite at, a position corresponding to R_c . Can the authors comment on the physical origin of this feature? Is this related to the persistence length/tumbling rate in relation to the value of τ chosen for the control? Is this accumulation at the edge a consequence of a finite delay in adjusting the velocity for outgoing cells? In fact, this peak does not seem to be present in the simulated data with instantaneous feedback shown in the SI. Am I right to think that this peak should be at a position relative to R_c of the order of the distance traveled over τ ?*

Authors: Indeed a peak appears at a close vicinity of R_c (which we now renamed to “a” in the manuscript for simplicity and cohesiveness). That originates from the discontinuity in velocities of bacteria going out of the confined circle experience, i.e., when inside the confined area, they have high speed v_1 , but if they escape their modulus suddenly becomes $v_0 < v_1$. Since the current of bacteria flowing out of the boundary must be continuous in the stationary state, the density must increase to compensate for a speed decrease. This doesn't happen for bacteria going in the direction of the active cloud, as they always move with maximum v_1 . This interpretation of the peak is also confirmed by numerical simulations, where there is no delay for velocity adjustment, and the peak is at R_c as in the experiments. The fact that in the previously reported simulations there was no central band in the axial case could have been misleading. We now add in the main text the new figure 3 that includes simulation results in the axial and radial case both showing the presence of a peak right at the edge of the uniformly illuminated region.

REVIEWERS' COMMENTS

Reviewer #1 (Remarks to the Author):

The authors have considerably improved the manuscript such that I recommend publication of the paper in Nature Comm in its present revised form.

Reviewer #2 (Remarks to the Author):

The authors have properly addressed all of my previous comments and those of the other referees. I find that the paper reads very nicely, and I recommend publication in its current form. The only issue is that the labels (b) and (c) in Fig. 3 have been inverted.

Reviewer #3 (Remarks to the Author):

I am satisfied with the revisions made by the authors, which have carefully addressed the comments of all reviewers and I am happy to recommend publication.

NCOMMS-22-02620B

“Rectification and confinement of photokinetic bacteria in an optical feedback loop”

by H. Massana-Cid, C. Maggi, G.Frangipane and R. Di Leonardo

Authors' reply to Reviewers comments

We thank all the referees for all the positive comments on our manuscript and for recommending publication. Our point-by-point responses and the corresponding changes to the manuscript are described below.

Response to Reviewer #1

Reviewer: *The authors have considerably improved the manuscript such that I recommend publication of the paper in Nature Comm in its present revised form.*

Authors: We thank the reviewer for helping us to improve our manuscript and for recommending publication in Nature Communications.

Response to Reviewer #2

Reviewer: *The authors have properly addressed all of my previous comments and those of the other referees. I find that the paper reads very nicely, and I recommend publication in its current form. The only issue is that the labels (b) and (c) in Fig. 3 have been inverted.*

Authors: We thank the reviewer for the positive feedback and for noticing the typo on the labels of Fig. 3. We correct it.

Response to Reviewer #3

Reviewer: *I am satisfied with the revisions made by the authors, which have carefully addressed the comments of all reviewers and I am happy to recommend publication.*

Authors: We thank the reviewer for helping us to improve our manuscript and for recommending publication in Nature Communications.